# Research on the Diagnosability of a Satellite Attitude Determination System on a Fault Information Manifold

**Ruotong Qu, Bin Jiang *** **and Yuehua Cheng**

College of Automation Engineering, Nanjing University of Aeronautics and Astronautics, Nanjing 210016, China
* Correspondence: binjiang@nuaa.edu.cn

**Abstract:** In this paper, a new method for fault diagnosability research based on information geometry is proposed. The problem of the diagnosability evaluation of dynamic system faults is transformed into a distance calculation problem on a manifold. The Fisher information distance is used to realize a quantitative judgment of diagnosability, and a quantitative evaluation index of the fault diagnosability of a satellite attitude determination system is designed. This includes a fault detectability index and a fault isolability index. The validity and superiority of the new indexes are verified through a mathematical simulation. In addition, the fault information is visually presented by the geodesics of the fault manifold, and the properties and behavior of the fault are mined and analyzed on the fault information manifold, which lays a foundation for further exploration of fault information through geometric methods.

**Keywords:** diagnosability; evaluation indexes; satellite attitude determination system; fault information manifold; Fisher information distance (FID)

## 1. Introduction

With the rapid development of aerospace technology, the importance of space in the civil and military fields has become increasingly prominent and, thus, has become a strategic fulcrum for competition among countries. Space technology is an essential basis for developing and utilizing space resources and safeguarding national security. Satellites are some of the most important tools for human exploration and space applications. Due to the high cost of satellite construction and the high cost of launching and in-orbit maintenance, once a failure occurs, it will cause considerable losses. In recent years, on-orbit fault diagnosis technology has made remarkable progress, which has effectively improved the on-orbit operation status and life of satellites. Over the years, studies on fault diagnoses of spacecraft have attracted extensive attention from scholars, and lots of valuable research results have emerged in both the theoretical and engineering fields [1–11]. Research such as that on fault detection and diagnosis [1–6], reconfigurability analysis [7–9], life or fault prediction [10–13], and fault tolerance [14–19] improves the security and reliability of satellite on-orbit operation from different perspectives.

Fault diagnosis is a significant means of achieving high-precision, long-life, and high-reliability on-orbit operation of satellite systems, and it is of great significance. However, existing research has mainly focused on the design of diagnostic algorithms, and there have been few studies on fault diagnosability. According to the definition of fault diagnosability [20], diagnosability includes detectability and isolability. Diagnosability is the key to fast and accurate fault detection and isolation in dynamic systems, and it is also the basis of the design of fault diagnosis methods. Without analyzing the diagnosability of faults, blindly designing fault diagnosis algorithms will inevitably consume personnel and material resources and may not achieve satisfactory results. Therefore, research on fault diagnosability is of great significance.

Most of the existing diagnosability research on general control systems only considers the situation of linear system faults [21–23]; research on nonlinear systems and nonlinear

faults can usually only achieve qualitative evaluation [24,25]. There have been few relatively studies on the diagnosability of satellite systems. For example, Liu [26] studied the fault diagnosability of a satellite momentum wheel based on the idea of if the transfer function from fault to output was zero and if the transfer functions between faults were different. Li and Wang used the distribution probability of fault vectors and the cosine similarity between different fault vectors in [27] to design a quantitative evaluation method for diagnosability based on directional similarity, which was verified by a satellite attitude determination system. In [28], the authors described a satellite control system as a class of affine nonlinear models by using the invariant minimum dual distribution to give the diagnosability evaluation, which included the qualitative and quantitative evaluation index of detectability and isolability.

This paper proposes a method of system diagnosability evaluation based on information geometry theory. The diagnosability problem of the system is described as a distance judgment problem of a multivariate distribution in statistics, and the Fisher information distance is used to quantify the diagnosability of system faults. A fault detectability index and fault isolation index are designed. The designed indexes have explicit and intuitive geometric significance, and they are real distance measures without the problem of index asymmetry. The principle and algorithm process of the method to realize diagnosability judgment are given. Taking the satellite attitude determination system as an example, the scientificity and effectiveness of the proposed method, as well as its superiority in analyzing nonlinear faults, are verified. The fault information and properties contained in the fault manifold geodesics are deeply excavated, and information geometry theory is used to analyze some essential problems in the process of fault development and evolution and to lay the foundation for future research on efficient fault detection, diagnosis, design, and optimization methods.

## 2. Mathematical Description of the Problem

A satellite attitude determination system is the basis for the attitude control of a satellite. Its task is to process information measured by the attitude sensors to obtain the attitude of the satellite body coordinate system relative to the orbit coordinate system [29,30]. The most commonly used attitude sensors include star sensors, sun sensors, Earth sensors, and gyroscopes [30,31].

A model of satellite dynamics can be equivalently transformed into three independent axes: the roll, pitch, and yaw. The pitch angular velocity of a satellite is a fast variable in comparison with the other two axes. In this paper, a satellite pitch-axis attitude determination system based on an "infrared Earth sensor + gyroscope" is taken as an example for analysis. The infrared Earth sensor and the gyroscope are used to measure the attitude angle and angular rate of the satellite, respectively, and the combined use of the two can achieve high-precision combined attitude determination. Considering that the attitude angle/angular velocity of the satellite's pitch axis (Y-axis) is decoupled from the roll and yaw axes (X-axis and Z-axis), and in order to reduce the model's dimensions and simplify the problem, the discrete-form state-space model of the satellite attitude determination system on the pitch axis is given below, as shown in Equation (1).

$$
\begin{cases}
\begin{bmatrix} \theta(k+1) \\ d_y(k+1) \\ b_y(k+1) \end{bmatrix} = \begin{bmatrix} 1 & -dt & -dt \\ 0 & 1-\frac{1}{\tau_y}dt & 0 \\ 0 & 0 & 1 \end{bmatrix} \begin{bmatrix} \theta(k) \\ d_y(k) \\ b_y(k) \end{bmatrix} + \begin{bmatrix} dt & 0 & 0 \\ 0 & dt & 0 \\ 0 & 0 & dt \end{bmatrix} \begin{bmatrix} \omega_0 + g_y \\ 0 \\ 0 \end{bmatrix} + \begin{bmatrix} dt & 0 \\ 0 & 0 \\ 0 & 0 \end{bmatrix} \begin{bmatrix} g_f \\ \theta_f \end{bmatrix} \\
\qquad\qquad + \begin{bmatrix} 1 & 0 & 0 \\ 0 & 1 & 0 \\ 0 & 0 & 1 \end{bmatrix} \begin{bmatrix} n_y(k) \\ n_{by}(k) \\ n_{dy}(k) \end{bmatrix} \\
\theta(k) = \begin{bmatrix} 1 & 0 & 0 \end{bmatrix} \begin{bmatrix} \theta(k) \\ d_y(k) \\ b_y(k) \end{bmatrix} + \begin{bmatrix} 0 & 1 \\ 0 & 0 \\ 0 & 0 \end{bmatrix} \begin{bmatrix} g_f \\ \theta_f \end{bmatrix} + n_\theta(k)
\end{cases} \tag{1}
$$

The mathematical model of the satellite attitude determination system of the infrared Earth sensor + gyroscope is expressed as above, where $\theta$ and $\omega_0$ represent the attitude angle and orbital angular velocity of the satellite, respectively; $g_y$ represents the pitch-axis angular rate of the satellite, which is the output of the gyroscope; $d_y$ and $b_y$ are the exponentially correlated drift and constant drift of the gyroscope that are caused by the small amount of torque interference on it; $\tau_y$ is the time constant of the gyroscope. $g_f$ stands for the fault vector, and $n_y/n_{by}/n_{dy}$ are the noises of the gyroscope's output and two drifts, respectively, which are all in the form of Gaussian white noise. $\theta_f$ and $n_\theta$ represent the fault vector and Gaussian white noise of the measured attitude angle output by the infrared Earth sensor, respectively. $dt$ represents the sampling time interval. The values of the related parameters are: $dt = 0.1$ s, and the time constant $\tau_y = 1/11$ (dimensionless); the orbital angular velocity $\omega_0 = 0.06$ rad/s, and the Gaussian white noises $n_y \sim N(0, 10^{-6})$, $n_{by} \sim N(0, 10^{-6})$, $n_{dy} \sim N(0, 10^{-6})$, $n_\theta \sim N(0, 10^{-6})$. Here, $\sim N$ is a Gaussian distribution symbol, and the distribution's parameters are shown in the parentheses.

Equation (1) can be simplified as follows:

$$\begin{cases} x(k+1) = Ax(k) + Bu(k) + B_f f_a(k) + B_w w(k) \\ y(k) = Cx(k) + Du(k) + D_f f_s(k) + D_v v(k) \end{cases} \tag{2}$$

In fact, state $x$, input $u$, and output $y$ may be influenced by the coupling of interferences $w$ and $v$, as well as by fault $f$. In order to separate the influence of state $x$ and decouple the fault from the interference, the following equation was constructed in [32].

$$Lz(k) = Hx(k+1) + Ff(k) + Ee(k) \tag{3}$$

where

$$z(k) = \begin{bmatrix} y(k) \\ u(k) \end{bmatrix}, f(k) = \begin{bmatrix} f_a(k) \\ f_s(k) \end{bmatrix}, e(k) = \begin{bmatrix} w(k) \\ v(k) \end{bmatrix},$$

$$L = \begin{bmatrix} 0 & -B \\ I & D \end{bmatrix}, H = \begin{bmatrix} A \\ C \end{bmatrix}, F = \begin{bmatrix} B_f & 0 \\ 0 & D_f \end{bmatrix}, E = \begin{bmatrix} B_w & 0 \\ 0 & D_v \end{bmatrix}$$

We pe-multiply both sides of the equation by the matrix $N_H$:

$$N_H Lz = N_H Ff + N_H Ee \tag{4}$$

where $N_H \bullet H = 0$, and $N_H Lz$ on the left side of the equation is the dynamic behavior of the system. Since the equivalent space transformation does not affect the solution of the system, Equation (4) can be used to describe the dynamic behavior of the attitude determination system shown in Equation (1). $N_H Ff$ on the right side of the equation is the fault vector, which is composed of the direction matrix $N_H F$ and the fault vector $f$. $N_H Ee$ is the interference vector and has a dynamic behavior. $N_H Lz$ is a multivariate distribution composed of the fault vector and the interference vector.

Therefore, the purpose of quantifying the detectability and isolability of the dynamic system faults shown in Equation (1) can be realized by measuring the similarity or difference in the multivariate distribution when no fault occurs or when different faults occur according to the corresponding criterion (such as the distance similarity or directional similarity).

## 3. Quantitative Diagnosability Evaluation Based on the Fisher Information Distance

Parameter vectors usually exist in an abstract manifold, and the manifolds corresponding to a real system usually have a complex topology. Consider the probability distribution parameter family $S = \{p(x|\theta)\}$, where $x$ is a random variable, $\theta = [\theta_1, \ldots, \theta_n]^T$ is an $n$-dimensional parameter vector with a particular distribution, and $S$ is a statistical manifold with the (local) coordinate system $\theta$. For a particular $\theta$ that belongs to the parameter space $\Theta \in R^n$, the measured and observed value of x belongs to the sampling space $X \in R^n$;

then, each $p(x|\theta)$ corresponds to an actual probability distribution, that is, each probability distribution $p(x|\theta)$ corresponds to a point on the statistical manifold S.

Take the vector $N_H Ff$ containing the fault information as the mean value $\mu(\theta)$ of the fault manifold and take the interference vector $N_H Ee$ as the variance $C(\theta)$ of the fault manifold, namely:

$$N_H Ff = N_H \begin{bmatrix} B_f & 0 \\ 0 & D_f \end{bmatrix} \begin{bmatrix} f(k) \\ f(k) \end{bmatrix} = N_H \begin{bmatrix} 0.1 & 0 & 0 & 0 \\ 0 & 0 & 0 & 0 \\ 0 & 0 & 0 & 1 \\ 0 & 0 & 0 & 0 \end{bmatrix} \tag{5}$$

$$N_H Ee = N_H \begin{bmatrix} B_w & 0 \\ 0 & D_v \end{bmatrix} \begin{bmatrix} w(k) \\ v(k) \end{bmatrix} = N_H \begin{bmatrix} w(k) \\ v(k) \end{bmatrix} \tag{6}$$

The values of coefficient matrices $B_f, D_f, B_w, D_v$ in Equation (1) are known; now, only $N_H$ needs to be determined, and $N_H$ is the left orthogonal basis for the null space of $H$, which means that $N_H H = 0$. The matrix $H$ is determined by the system matrix $A$ and the output matrix $C$ of Equation (1) [32]. Extending the system to a full-dimensional observable system, let $C = I_3$; then,

$$H = \begin{bmatrix} A \\ C \end{bmatrix} = \begin{bmatrix} 1 & -0.1 & -0.1 \\ 0 & -0.990909 & 0 \\ 0 & 0 & 1 \\ 1 & 0 & 0 \\ 0 & 1 & 0 \\ 0 & 0 & 1 \end{bmatrix} \tag{7}$$

Then,

$$\mu(\theta) = \begin{bmatrix} -0.0705 & 0.7050 \\ 0.0015 & -0.0148 \\ 0.0015 & -0.0146 \end{bmatrix} \begin{bmatrix} \arctan f_{gy}(k) \\ \sqrt{f_{hy}(k)} \end{bmatrix} = \begin{bmatrix} -0.0705 \arctan f_{gy}(k) + 0.705\sqrt{f_{hy}(k)} \\ 0.0015 \arctan f_{gy}(k) - 0.0148\sqrt{f_{hy}(k)} \\ 0.0015 \arctan f_{gy}(k) - 0.0146\sqrt{f_{hy}(k)} \end{bmatrix} \tag{8}$$

$$C(\theta) = \begin{bmatrix} N\left(0, 10^{-4}\right) & 0 & 0 \\ 0 & N\left(0, 10^{-4}\right) & 0 \\ 0 & 0 & N\left(0, 10^{-4}\right) \end{bmatrix} \tag{9}$$

Different values of the parameter $\theta$ represent different types of faults. Considering the nonlinearity caused by the sensor measurement conversion, the settings in this paper are as follows:

$$\theta = f^T(k) = \begin{bmatrix} g_f & \theta_f \end{bmatrix} = \begin{bmatrix} \arctan f_{gy}(k) & \sqrt{f_{hy}(k)} \end{bmatrix} \tag{10}$$

The manifold of a fault system is obtained, and the fault probability distribution $p(x|\theta)$ and parameterized probability distribution family $S = \{p(x|\theta)\}$ constitute an n-dimensional statistical manifold. Here, $\theta$ is the distribution parameter (and the fault parameter vector), $\mu(\theta)$ is the mean value, and $C(\theta)$ is the variance. Different faults have different $\mu(\theta)$ and $C(\theta)$ in this manifold. In this statistical manifold, $\theta$ is the coordinate system and is global.

In statistical manifolds, a Fisher information matrix (FIM) is the unique Riemannian geometric metric tensor for the parameterized probability distribution family [33], and it is expressed as $(G(\theta) = [g_{\alpha\beta}(\theta)]$. The FIM is given by the following:

$$g_{\alpha\beta}(\theta) = E\{\frac{\partial \log p(x|\theta)}{\partial \theta_\alpha}, \frac{\partial \log p(x|\theta)}{\partial \theta_\beta}\} \tag{11}$$

where $E$ is the mathematically expected value. As the parameter $\theta'$ approaches $\theta$, FIM measures the ability to distinguish between two adjacent parameters $\theta'$ and $\theta$ from the data $x$. This equation can be rewritten in parametric form as:

$$g_{\alpha\beta}(\theta) = \left[\frac{\partial\mu(\theta)}{\partial\theta_\alpha}\right]^T C^{-1}(\theta)\left[\frac{\partial\mu(\theta)}{\partial\theta_\beta}\right] + 0.5 * tr\left[C^{-1}(\theta)\frac{\partial C(\theta)}{\partial\theta_\alpha} * C^{-1}(\theta)\frac{\partial C(\theta)}{\partial\theta_\beta}\right] \quad (12)$$

According to the Equation (12), the information metric $g_{\alpha\beta}(\theta)$ of the attitude determination system studied under the fault set is Equation (13):

$$g_{\alpha\beta}(\theta) = \begin{bmatrix} \dfrac{19{,}899}{400\left(1+f_{gy}{}^2(k)\right)^2} & -\dfrac{248{,}733}{1000\left(1+f_{gy}{}^2(k)\right)\sqrt{f_{gy}}} \\ -\dfrac{248{,}733}{1000\left(1+f_{gy}{}^2(k)\right)\sqrt{f_{hy}(k)}} & \dfrac{1{,}243{,}643}{1000f_{hy}(k)} \end{bmatrix} \quad (13)$$

Its determinant is:

$$\frac{49{,}707}{2{,}000{,}000\left(1 + f_{gy}{}^2(k)\right)^2 f_{hy}(k)} \quad (14)$$

On a manifold, since space is curved, to define the distance between two points on the manifold, the length of the curve connecting the two points on the manifold should first be defined.

The differential distance between two points (or two distributions) $p(x|\theta)$ and $p(x|\theta + d\theta)$ on a manifold can be expressed by the metric:

$$ds^2 = \Sigma_{\alpha\beta}g_{\alpha\beta}d\theta_\alpha d\theta_\beta = d\theta^T G(\theta)d\theta \quad (15)$$

Considering that $\theta(t) \in \Theta$ is the curve connecting $\theta_1 = \theta(t_1)$ and $\theta_2 = \theta(t_2), t_1 \leq t \leq t_2$, this curve can be described as a parametric equation with a single free parameter $t$. The distance between the distributions $p(x|\theta_1)$ and $p(x|\theta_2)$ on the statistical manifold is defined by the curve $\theta(t)$ [34]:

$$D_F(\theta_1, \theta_2) = \int_{t_1}^{t_2}\left(\sqrt{\left(\frac{d\theta}{dt}^T\right)G(\theta)\frac{d\theta}{dt}}\right)dt \quad (16)$$

where $\triangleq$ stands for "define as". This integral distance depends on the choice of the curve $\theta(t)$. Generally, the minimum value of curves for all possible connections is defined as the integral distance between the two distributions, and it is called the Fisher information distance (FID). The Fisher information distance between the distributions $p(x|\theta_1)$ and $p(x|\theta_2)$ is expressed as [35]:

$$D_F(\theta_1, \theta_2) = \min_{\theta(t):\theta(t_1)=\theta_1, \theta(t_2)=\theta_2}\int_{t_1}^{t_2}\left(\sqrt{\left(\frac{d\theta}{dt}^T\right)G(\theta)\frac{d\theta}{dt}}\right)dt. \quad (17)$$

The curve with the smallest distance defined above is actually the geodesic that connects two points on the manifold, while the Fisher information distance is the length of the shortest geodesic connecting the two points. Geodesics are generalizations of straight lines on manifolds in Euclidean space, and the Fisher information distance is a generalization of the Euclidean distance on manifolds.

The FID satisfies the property of the distance definition, and it is symmetrical for all $\theta_1, \theta_2 \in \Theta$:

$$D_F(\theta_1, \theta_2) = D_F(\theta_2, \theta_1) \quad (18)$$

For $\theta_1, \theta_2, \theta_3 \in \Theta$, the FID satisfies the triangle inequality:

$$D_F(\theta_1, \theta_2) + D_F(\theta_2, \theta_3) \geq D_F(\theta_1, \theta_3) \quad (19)$$

Based on the above studies, the following detectability evaluation index is designed:

$$F_D(f_i) = D_F(\theta_i, \theta_0) \tag{20}$$

where $\theta_0$ represents the parameters of the fault manifold in the normal state or fault-free state, and $\theta_i = f_i(k)$, $i = 1, 2, \ldots n$, represents the parameters of the fault manifold at time $k$ in the case of a fault $f_i$. $g_{\alpha\beta}(\theta)$ is an all-zero matrix in the case of the fault-free state.

The following isolability evaluation index is designed:

$$F_I\left(f_i, f_i'\right) = D_F\left(\theta_i, \theta_i'\right) \tag{21}$$

where $\theta_i$ and $\theta_i'$ are the parameters of different fault manifolds $f_i$ and $f_i'$, respectively. By measuring the difference in the FIDs of different faults, different faults can be separated. It is necessary to notice that $F_{iso}\left(f_i, f_i'\right) = F_{iso}\left(f_i', f_i\right)$, the isolability index is symmetrical.

The definition of the Fisher information distance given by Equation (17) requires solving the minimum value of the integral, which is a variational problem whose solution is given by the geodesic equation.

$$\frac{d^2\theta^v}{dt^2} + \Gamma^v_{\alpha\beta}\frac{d\theta_\alpha}{dt}\frac{d\theta_\beta}{dt} = 0, \ \forall v \in \{1, \ldots, n\} \tag{22}$$

Einstein's summation convention is used in the above formula, where $\theta(t) = [\theta^1(t), \ldots, \theta^n(t)]^T$ is the coordinate of the geodesic $C\gamma(t)$, and $\Gamma^v_{\alpha\beta}$ is the Christoffel symbol of the second kind [36] in this coordinate system, which is usually used to represent a class of Riemannian connection coefficients, which can be obtained with the Fisher information matrix according to the following formula:

$$\Gamma^v_{\alpha\beta} = \frac{1}{2}g^{v\lambda}\left(g_{\lambda\alpha,\beta} + g_{\beta\lambda,\alpha} - g_{\alpha\beta,\lambda}\right) \tag{23}$$

where $[g^{v\lambda}]$ is the coordinate component of the inverse matrix of the Fisher information matrix $G(\theta) = [g_{\alpha\beta}(\theta)]$, which can be obtained directly through inversion.

The geodesic equation in Equation (9) is an ordinary differential equation of the coordinate $\theta(t)$. For the given initial conditions $\theta(0)$ and $\dot{\theta}(0)$, the solution of the geodesic equation is unique.

$$T^b\nabla_b T^a = \theta \tag{24}$$

The above equations are called geodesic equations, and there is a metric tensor field $g_{ab}$ on the manifold $S$; the geodesic of the manifold $(S, g_{ab})$ is the geodesic of $(S, a)$, where $\nabla_a$ fits with $g_{ab}$ ($\nabla_a g_{ab} = 0$).

$\theta^v = \theta^v(t)$ is the parametric form of the geodesic $C\gamma(t)$; then, the above equation can be rewritten into the coordinate component form, as shown in Equation (22). The geodesic Equation (24) is an ordinary differential equation of the coordinate $\theta(t)$. For the given initial conditions $\theta(0)$ and $\dot{\theta}(0)$, the solution of the geodesic equation is unique.

The Fisher information distance is the length of the shortest geodesic connecting two points (i.e., two distributions) on a manifold. Geodesics connected to different endpoints also express the detectability and separability of different faults.

Therefore, according to Equation (17), the FID can be obtained after integration, and it can be used to measure the distance on a manifold between different distributions that represent a fault-free state, a fault, and different faults so as to achieve a quantitative evaluation of diagnosability.

The corresponding non-zero Christoffel symbol of the second kind is:

$$\Gamma^1_{1,1} = -\frac{2f_{gy}(k)}{1 + f_{gy}{}^2(k)} \tag{25}$$

$$\Gamma_{2,2}^2 = -\frac{1}{2f_{hy}(k)} \tag{26}$$

The corresponding geodesic equation of the attitude determination system is:

$$\begin{cases} f''_{gy}(k) = \frac{2f_{gy}(k)[f'_{gy}(k)]^2}{1+[f'_{gy}(k)]^2} \\ f''_{gy}(k) = \frac{[f'_{hy}(k)]^2}{2f_{hy}(k)} \end{cases} \tag{27}$$

## 4. Simulation Experiment

Summarizing the content in Section 2, the algorithm for the quantitative evaluation method for diagnosability is provided in Algorithm 1:

---

**Algorithm 1:** The quantitative evaluation method for diagnosability on an information manifold

---

**Require**: State–Space Model of a Dynamic System:
1. Pretreatment: $Lz(k) = Hx(k+1) + Ff(k) + Ee(k)$
2. Pretreatment: $N_H Lz = N_H Ff + N_H Ee$
3. Manifold parameter: Mean value $\mu(\theta) = N_H Ff$, Variance $C(\theta) = N_H Ee$
4. Fisher Information Metric $G(\theta) = g_{\alpha\beta}(\theta) = f(\mu(\theta), C(\theta))$
5. Fisher Information Distance (FID) $D_F(\theta_1, \theta_2) = \int_{t_1}^{t_2} \left( \sqrt{\left(\frac{d\theta}{dt}^T\right) G(\theta) \frac{d\theta}{dt}} \right) dt$
6. Detectability Index $F_D(f_i) = D_F(\theta_i, \theta_0)$
7. Isolability Index $F_I\left(f_i, f'_i\right) = D_F\left(\theta_i, \theta'_i\right)$
8. Import specific fault/faults to obtain the detectability/isolability index

---

To verify the effectiveness of the proposed algorithm, in this section, for the satellite pitch-axis attitude determination system, which is shown in Equation (1), a simulated experiment on diagnosability evaluation is carried out in a joint mathematical and Matlab simulation platform.

As the derived manifold parameters $\mu(\theta), C(\theta)$ already contain the information of the satellite pitch-axis attitude determination system (Equation (1)), as shown in Equation (10), $f_{gy}$ is the fault vector of the gyroscope, and $f_{hy}$ is the fault vector of the infrared Earth sensor. Each fault vector contains a nonlinearity related to its sensor output characteristics.

We set the initial value to $f_{gy}(0) = 0, f_{gy}'(0) = 0.1, f_{hy}(0) = 1, f_{hy}'(0) = 0.1$. The time-varying geodesics for the two fault components are obtained, as shown in Figure 1.

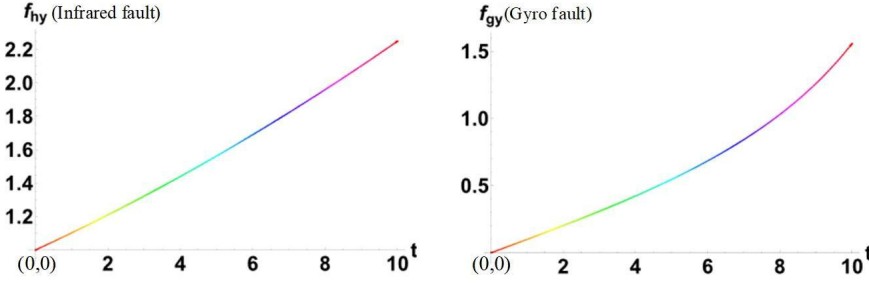

**Figure 1.** Development of single faults $f_{gy}$ and $f_{hy}$ on a fault information manifold over time.

The FID of $f_{gy}(k)$ is 1.55741, and the FID of $f_{hy}(k)$ is 1.25, which means that

$$F_D\left(f_{gy}\right) = 1.55741$$

$$F_D\left(f_{hy}\right) = 1.25$$

It can be found that the geodesic images of the $f_{gy}(k)$ fault component and the $f_{hy}(k)$ fault component also reflect this difference; the derivatives of the geodesics of the two fault components are different. It should be noted that the two geodesics in Figure 1 only reflect the development and variation of a single fault itself on the manifold.

For the two-dimensional fault information manifold (including two fault components) studied in this paper, there are three possible fault types: two single-fault cases and one compound fault case.

A fault information manifold with a curvature is a typical surface that is hard to display. To intuitively demonstrate its shape and study its properties, this paper then "transforms the surface into a plane" through geodesics, then researches the fault diagnosability problem on it. In this paper, by taking the two fault components as the X-axis and Y-axis, respectively, the two-dimensional fault manifold surface can be mapped in the Cartesian coordinate system. In this coordinate system, the X-axis ($f_{gy}$) and Y-axis ($f_{hy}$) represent two single-fault cases (a gyroscope fault and infrared sensor fault), while the first quadrant represents a compound fault case. In the same type of fault case, different points in the coordinate system indicate a fault with different parameters.

Plotting the geodesics, the figure shows the path taken by a group of composite fault geodesics with a departure velocity of 1 that are located at the fault manifold coordinate (4,4) in the same time period. The compound fault parameters are set as follows: $f_{gy}(0) = 4, f_{hy}(0) = 4, f'_{gy}(0) = 1 * \sin[2\pi * \frac{j}{MX}], fhy'(0) = 1 * \cos[2\pi * \frac{j}{MX}]$. We set the number of geodesic lines to MX = 128.

This set of geodesics describes the unit circle on the fault information manifold with the same FID centered at the location of the compound fault, the point (4,4). The FID unit circle of a statistical manifold does not usually correspond to a circle in a Euclidean space, and vice versa. In fact, on the fault information manifold, the endpoints of these 128 geodesics form a "unit circle", but when mapped in Euclidean space, the shape of this unit circle is distorted, forming a shape similar to that of a "comet". Figure 2 also expresses the response of the FID to the fault development process in this fault form.

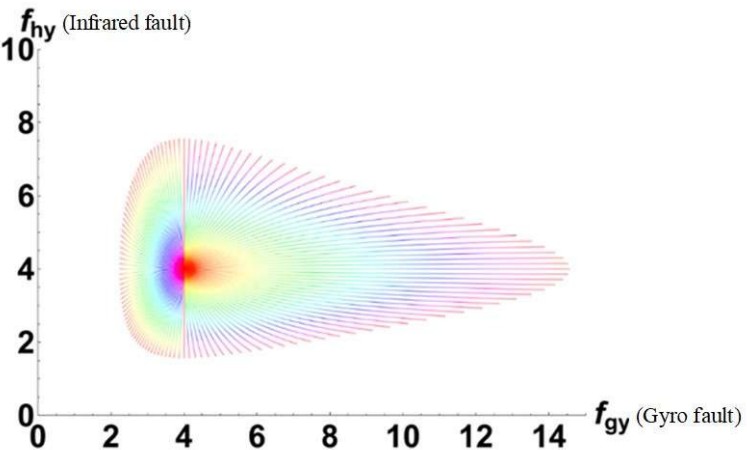

**Figure 2.** The FID unit circle of the compound fault $f_1$.

Because the two single faults are set to the x and y rectangular coordinate axes, the manifold, which was originally a curved surface, is "leveled", and the FID, which was originally a unit circle on the manifold, is distorted in this process. In this coordinate system, the evaluation of the diagnosis of faults based on the FID can be succinctly and intuitively described and studied.

As mentioned above, the X-axis ($f_{gy}$) and Y-axis ($f_{hy}$) represent two single-fault scenarios, and the first quadrant represents a compound fault scenario. Now, three types of faults are displayed in the coordinate system: a single gyro fault $f_{gy}$, a single infrared sensor fault $f_{hy}$, and a compound fault $f_1$:(4,4).

According to the definition of detectability in this paper, the detectability indexes of single faults $f_{gy}$ and $f_{hy}$ and of $f_1$ are shown in Figure 3, indicating FID of the line between the coordinates where the fault is located and the origin. It should be noted that the "line" here is actually the geodesic on the manifold, not a line mapped in Euclidean space.

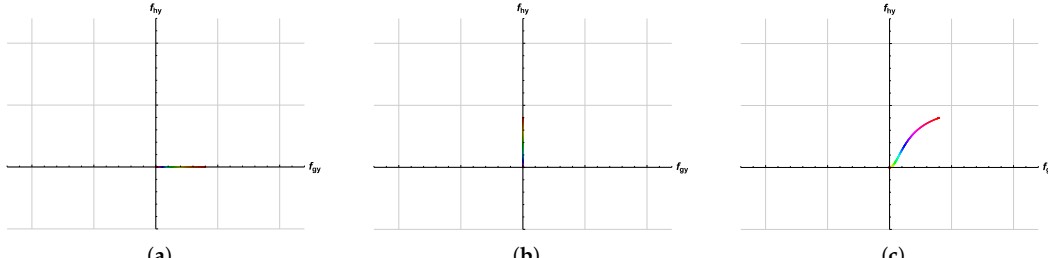

(a)  (b)  (c)

**Figure 3.** Fault detectability indexes on the fault information manifold. (**a**) $F_D(f_{gy})$; (**b**) $F_D(f_{hy})$; (**c**) $F_D(f_1)$.

The isolability between the single faults $f_{gy}$ and $f_{hy}$ and the compound fault $f_1$ is shown in the Figure 4, indicating the FIDs of the geodesics between the two faults. The line shown in Figure 4a is obviously not the shortest line between two points in Euclidean space, but it is the shortest path between them on a manifold.

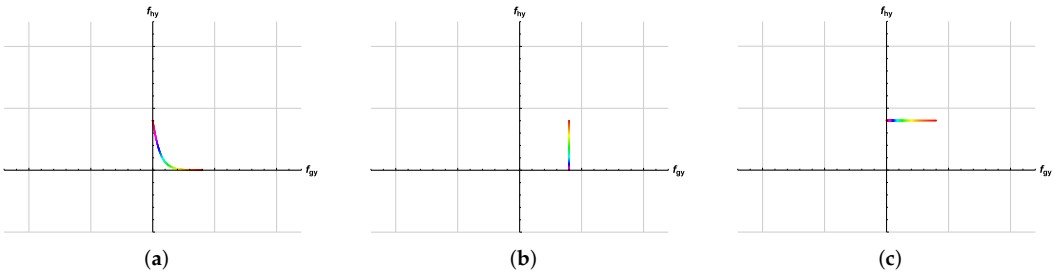

(a)  (b)  (c)

**Figure 4.** Fault isolability indexes on the fault information manifold. (**a**) $F_I(f_{gy}, f_{hy})$; (**b**) $F_I(f_1, f_{gy})$; (**c**) $F_I(f_1, f_{hy})$.

Table 1 shows the numerical results of the detectability and isolability indexes for the three types of faults.

**Table 1.** Diagnosability indexes on the fault information manifold for the three types of faults.

| Fault | $F_D$ | $F_I(, f_{gy})$ | $F_I(, f_{hy})$ | $F_I(, f_1)$ |
|---|---|---|---|---|
| $f_{gy}$ | 9.351268 | 0 | 9.351245 | 150.237868 |
| $f_{hy}$ | 141.061220 | 9.351268 | 0 | 141.061290 |
| $f_1$ | 131.709952 | 150.237868 | 141.061290 | 0 |

Firstly, the coordinates of the compound fault $f_1$ under the fault information manifold can be expressed as (9.351245, 141.061290), indicating that the isolability between the compound fault and $f_{hy}$ (single infrared fault) is stronger than that of $f_{gy}$ (single gyro fault). It can also be stated that the coupling between $f_{gy}$ and the compound fault $f_1$ is deep and difficult to isolate.

Secondly, it can be found that $F_I$ on the right is a symmetric matrix. This is because the FID is a true distance measure with symmetry, and it is fundamentally more scientific and accurate than diagnosability evaluation methods based on the KLD. The principle of the diagnosability evaluation method based on the KLD is stated briefly as follows.

The distance between two distributions (in this paper, two faults) $p(x|\theta_i)$ and $p(x|\theta_j)$ can be approximated with various alternative approximations. A common alternative to the information distance is the KLD (Kullback–Leibler divergence).

$$
\begin{aligned}
KLD[p(x|\theta_i)||p(x|\theta_j)] &= \int p(x|\theta_i) \ln \frac{p(x|\theta_i)}{p(x|\theta_j)} dx \\
&= E\{\ln p(x|\theta_i) - \ln p(x|\theta_j)\}
\end{aligned}
\tag{28}
$$

The relationship between the KLD and differential Fisher information distance is:

$$
ds^2 = 2KLD[p(x|\theta)||p(x|\theta + d\theta)]
\tag{29}
$$

The KLD provides a measure of the distance between two points on a manifold, but the KLD cannot give the shortest path between two points, which means that the KLD does not contain information about the structure of the manifold. This is also one of the most significant differences between the KLD and the Fisher information distance. At the same time, the KLD is not a true distance measure because it does not satisfy the symmetry and triangle inequality of the distance definition.

Since the KLD of the two distributions is equivalent to the maximum likelihood estimate between them, it can be used to evaluate the detectability of faults $f_i$ and the isolability between faults $f_i$ and $f_j$, which are calculated as follows:

$$
Detectability(f_i) = KLD[p(x|\theta_i)||0]
\tag{30}
$$

$$
Isolability(f_1, f_2) = KLD[p(x|\theta_1)||p(x|\theta_2)]
\tag{31}
$$

The numerical results of the detectability and isolability indexes of the three types of faults based on the KLD with the same settings are shown in the following.

It can been seen from Table 2 that though the diagnosability indexes obtained with the KLD can realize the evaluation of fault detectability and isolability, the evaluation value of each index appears to be chaotic and irregular. Because the KLD is asymmetric, the value of the isolability evaluation between $f_i$ and $f_j$ is unequal to the value of the isolability evaluation between $f_j$ and $f_i$. The problem of asymmetry in the fault isolability index is inevitable for a diagnosability evaluation based on the KLD [27]. Similarly, the diagnosability evaluation method based on the Bhattacharyya coefficient (BC) has the same problem [37]. The FID is a significant concept in information and statistical theory. It has made some achievements in theoretical and applied research on signal processing, target tracking, path planning, and other fields. In the field of fault diagnosis, the results obtained with the diagnosability method proposed in this paper have a similar tendency to that of the results obtained with the traditional KLD method, which demonstrates their correctness on the other side. At the same time, the method presented in this paper contains accurate fault information, and there is no problem of asymmetry in the isolability index.

**Table 2.** Diagnosability indexes based on the KLD for the three types of faults.

| Fault | *Det* | *Iso(, $f_{gy}$)* | *Iso(, $f_{hy}$)* | *Iso(, $f_1$)* |
|---|---|---|---|---|
| $f_{gy}$ | 43.722891 | 0 | 43.209558 | 11,304 |
| $f_{hy}$ | 9949.144 | 47.283433 | 0 | 9931.571 |
| $f_1$ | 8673.768471 | 11,312.5 | 9796.349 | 0 |

However, for the method proposed in this paper, the diagnosability problem is transferred to a fault information manifold for research, and the real distance measurement of the FID is used to design detectability and isolability indexes, so there is no such problem of asymmetry. The design of the indexes with the FID is relatively more scientific and is conducive to the development of subsequent research on diagnosability (such as fault diagnosis, diagnosability optimization, etc.).

Otherwise, through comparison with Table 1, we find that there is a certain relationship between the diagnosability evaluation values based on the FIDs of the three types of faults. It is manifested as: The isolability evaluation value of the single faults $f_{gy}$ and $f_{hy}$ is equal to the sum of the detectability evaluation values of $f_{gy}$ and $f_{hy}$.

$$F_D(f_{gy}) + F_D(f_{hy}) = F_I(f_{gy}, f_{hy}) = F_I(f_{hy}, f_{gy}) \tag{32}$$

The detectability evaluation value of the compound fault $f_1$ is equal to the modulus of the difference between the two detectability evaluation values of the single faults $f_{gy}$ and $f_{hy}$.

$$|F_D(f_{gy}) - F_D(f_{hy})| = F_D(f_1) \tag{33}$$

For verification, we set multiple compound faults with different parameters.

From Table 3, it can be noticed that in the new cases, there is indeed some relationship between the diagnosability indexes based on the FID, despite the numeral error that exists because of the limits of decimal digits. However, the diagnosability indexes obtained with the KLD do not have this connection.

**Table 3.** Diagnosability indexes on a fault information manifold for compound faults.

| Fault | $F_D$ | $F_I(, f_{gy})$ | $F_I(, f_{hy})$ |
|---|---|---|---|
| $f_2(4, 3)$ | 112.811393 | 121.653057 | 9.351245 |
| $f_3(1, 1)$ | 64.618801 | 70.140214 | 5.539564 |
| $f_4(1, 3)$ | 116.118182 | 121.653057 | 5.539564 |
| $f_5(3, 1)$ | 61.390573 | 70.140216 | 8.809759 |
| $f_6(3, 7)$ | 177.282673 | 186.091232 | 8.809759 |
| $f_7(5, 5)$ | 148.024455 | 157.201838 | 9.686858 |
| $f_8(5, 9)$ | 201.904966 | 211.050089 | 9.686858 |
| $f_9(7, 5)$ | 147.632920 | 157.201838 | 10.078330 |
| $f_{10}(9, 7)$ | 176.307810 | 186.091233 | 10.298639 |

The four representative geodesic lines in Figure 2 are selected and displayed separately.

It can be seen in Figure 5 that among these geodesic lines (there are theoretically infinite ones), there are two special geodesic lines, which are the shortest geodesic lines in the "unit circle"; in addition, the FIDs of the two special geodesic lines are equal, and the other geodesic lines are symmetrical about these two special geodesic lines. According to the research in [38], they are named the "symmetry lines", and "symmetry lines" have extremely important properties in the information space.

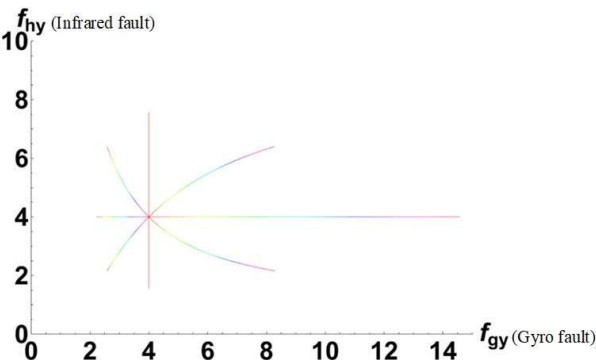

**Figure 5.** Representative geodesics of $f_1$ on the fault information manifold.

In the figure, the two geodesic "symmetry lines" appear to be "straight lines" and have unequal lengths because they can only be displayed through the Euclidean plane. However, with the Riemann metric, these two special geodesics are both typical curves and

have the same length. According to [38], a "symmetry line" embodies a certain symmetry and conservation principle in the information space.

With the Riemannian metric, free particles travel unequal distances along different geodesics to accumulate the same amount of energy. The geodesics between two points in a Riemannian manifold are not unique, and the FID between two points corresponds to the geodesics of the shortest length. Since on different geodesics, free particles have the same energy that is accumulated in a certain period of time, according to the "principle of lowest energy", free particles must choose to travel the shortest path. In the information space, the geodesics corresponding to the FID also correspond to the shortest path for accumulating energy from the "initial state" endpoint to the "final state" endpoint, which can also be regarded as the inevitable path of information. For the fault information space, after a certain fault occurs in the system, this state can also be understood as an "initial state". In the absence of the injection of new faults, the "initial state" will follow the geodesic line and move to reach a certain "final state". The geodesic corresponding to the FID reflects the developmental trajectory of the fault.

**Remark 1.** *1. There are special geodesic "symmetry lines" that exist on a fault information manifold, and they can represent the diagnosability properties of a fault. A "symmetry line" is an inevitable path for the development and evolution of a fault after setting the "initial state".*
*2. There is a special geodesic "symmetry line" of a fault component whose length reflects how detectable the fault is. Since the geodesic is the path that the fault travels on the manifold with the same departure speed and the same time interval, a longer path means a richer amount of information, which is more beneficial for researchers' measurement work.*
*3. The faults studied in this paper are coupled with two faults, and the coupling between the faults causes a deformation of the FID unit circle, distorting the unit circle into a "comet-like" shape. The closer the fault information is to the "comet tail", the easier it is to decouple, and the longer the "comet tail" is, the easier it is for the fault to be decoupled. This effect is now called the "comet tail effect" on the fault manifold.*

## 5. Conclusions

To realize the reliable in-orbit operation of satellites and improve the fault diagnosis capabilities of their systems in the design stage, a fault diagnosability evaluation method based on information geometry is proposed in this paper. The Fisher information distance is used to realize the quantitative evaluation of the fault diagnosability of a satellite's attitude determination system. The proposed fault diagnosability evaluation method is independent of specific fault diagnosis schemes, fault nonlinearities, and system interferences, and it is suitable for multi-fault situations. The designed diagnosability indexes have explicit and intuitive geometric significance, and they solve the problem of the asymmetry of the fault isolability indexes designed with the traditional diagnosability algorithm based on the distance similarity. Fault information is expressed "geometrically" through the geodesics of the fault manifold; this may inspire efficient fault detection, diagnosis, design, and optimization methods.

**Author Contributions:** Methodology, validation, and writing—original draft preparation, R.Q.; writing—review, editing, and supervision, Y.C. and B.J.; funding support, B.J. and Y.C. All authors have read and agreed to the published version of the manuscript.

**Funding:** This research was funded by the National Natural Science Foundation of China (62003162) and the Fundamental Research Funds for the Central Universities (NZ2020003).

**Institutional Review Board Statement:** Not applicable.

**Informed Consent Statement:** Not applicable.

**Data Availability Statement:** Not applicable.

**Conflicts of Interest:** The authors declare no conflict of interest.

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
