# Peer review of "Research on the Diagnosability of a Satellite Attitude Determination System on a Fault Information Manifold"

_applsci, doi:10.3390/app122412835_

Round 1

Reviewer 1 Report

The paper discusses the diagnosability of satellite attitude determination systems. The topic is of great interest, but the description is very cryptic, with few applicative correlations to the spacecraft attitude determination scenario. The method is mainly theoretical, and the technological or scientific application is hidden behind the method and its discussion. It is hard to correlate the proposed analyses and the associated results to the faults that can affect an attitude determination subsystem. Moreover, the authors do not clearly distinguish their findings from the ones available in literature. The current version of the manuscript does not allow to understand the method, its benefit to the proposed applicative scenario and the extent of its performances and validity. The paper shall undergo a deep revision, in order to be acceptable for publication. 

Specific comments follows:

- Introduction shall be numbered as section 1, and not 0.

- Please, refer to reference works by stating the name of the authors (at least the main one) in the text. Please, avoid: "the paper [xx] studies ...", "[xx] used ..." and so on.

- Literature review does not consider many recent works on the topic, and it is very focused on just national references, not very diverse at international level. For example, refer to the following and not exhaustive list of relevant works:

    - Hasan, Muhammad Noman, Muhammad Haris, and Shiyin Qin. "Fault-tolerant spacecraft attitude control: A critical assessment." Progress in Aerospace Sciences 130 (2022): 100806.

    - Morales-Reyes, Alicia, et al. "Fault tolerant and adaptive gps attitude determination system." 2009 IEEE Aerospace conference. IEEE, 2009.

    - Colagrossi, Andrea, and Michèle Lavagna. "Fault Tolerant Attitude and Orbit Determination System for Small Satellite Platforms." Aerospace 9.2 (2022): 46.

- P2L62 "Attitude measurement components" are typically referred as "sensors". Please, specify what you mean with that definition.

- P2L64 "Commonly used measurement components include: star sensors, sun sensors, earth sensors and gyroscopes." Please, add a reference.

- P2L65 "infrared earth sensor + gyroscope" is a set of sensors that does not allow full attitude estimation. Please, comment on this choice and state the applicability ranges. Moreover, when you refer to "attitude angle and angular rate" you should link this output to the selection of attitude sensors.

- Eq. 1 seems a set of 3 equations. But it is not. Please, reformat this equation and better introduce it. Moreover, the discretisation of the equation is not clearly stated, but only desumed from the form (e.g. k and k+1 terms). Please, define k and extend the description of this equation.

- P2L68 Please, better clarify the association between the symbols and what they are. Do not list all the symbols and, then, all the descriptions.

- P3L72 Units shall be always stated (even if adimensional), and the Gaussian distribution symbol (desumed) shall be clearly explained.

- Section 2. Please, extend the description and the discussion on the method, adding relevant references. Moreover, please justify the selection of the numerical values, and of the assumed distribution. Clearly state it in the text.

- Please, clearly distinguish your contribution from existing literature works.

- Section 3. Also in this section the experiments shall be motivated, and all the numerical values need to be justified. Add a introduction describing  what is the simulation environment and why it has been selected. The current description is very cryptic and the connection with the attitude determination problem and the attitude dynamics is lost. The authors are invited to make direct explicit link between the diagnosability discussed in this paper and the fault detection that can be practically executed on-board a spacecraft.

- Section 3. All the results are commented describing the numerical values and the plots. It is almost impossible to maintain a connection with the spacecraft application and the fault diagnosis output. Moreover, all the numbers are not related to a realistic case (e.g. orbit, sensors performances, attitude dynamics, spacecraft class, ...). The authors invited to deeply revise the presentation of their results.

- Conclusions are not clearly supported by the presented results.

Reviewer 2 Report

This paper proposed a new method for fault diagnosability of the Satellite Attitude Determination research based on information geometry. The description is clear and contributions are explicit. This paper can be accepted after a minor revision.

1.      The contributions should be highlighted in Introduction.

2.      The flow chart of the proposed method should be provided.

3.      Name of coordinates in figures should be clearer.

4.      What is the difference of the proposed method and the traditional method, such as FMD proposed in “Feature mode decomposition: new decomposition theory for rotating machinery fault diagnosis” or “Practical framework of Gini index in the application of machinery fault feature extraction”.

Round 2

Reviewer 1 Report

All my previous comments were satisfactorily answered.